# VLMs have Tunnel Vision: Evaluating Nonlocal Visual Reasoning in Leading VLMs

**Shmuel Berman**    **Jia Deng**
Princeton University, Department of Computer Science
{sb6870, jiadeng}@princeton.edu

## Abstract

Vision-Language Models (VLMs) excel at complex visual tasks such as VQA and chart understanding, yet recent work suggests they struggle with simple perceptual tests. We present an evaluation that tests vision-language models' capacity for *nonlocal visual reasoning*—reasoning that requires chaining evidence collected from multiple, possibly distant, regions of an image. We isolate three distinct forms of nonlocal vision: *comparative perception*, which demands holding two images in working memory and comparing them; *saccadic search*, which requires making discrete, evidence-driven jumps to locate successive targets; and *smooth visual search*, which involves searching smoothly along a continuous contour. Flagship models (e.g. GPT-5, Gemini 2.5 Pro, Claude Sonnet 4), even those that perform well on prior primitive-vision benchmarks, fail these tests and barely exceed random accuracy on two variants of our tasks that are trivial for humans. Our structured evaluation suite allows us to test if VLMs can perform similar visual algorithms to humans. Our findings show that despite gains in raw visual acuity, current models lack core visual reasoning capabilities.

## 1 Introduction

Vision-Language Models (VLMs) have demonstrated impressive performance on complex multimodal tasks. These models appear to possess a deep understanding of images and achieve over 90% accuracy on benchmarks like AI2D and ChartQA [10, 14, 12]. Such high-level competence would suggest these models have strong primitive visual skills. However, recent work such as VLMs are Blind [16] or HallusionBench [6], shows that VLMs excel at answering high-dimensional questions that require background context but fail at recognizing simple geometry. Yet newer models score higher on these adversarial benchmarks, indicating that they possess stronger visual acuity.

While higher performance indicates stronger visual perception, these adversarial benchmarks do not test sequential visual reasoning. Most tasks can be simulated as a series of independent extractions from the image-space to text-space, where each look at the image content is independent and does not *require* understanding relationships within the image itself. The actual reasoning can then occur in text space, where Large Language Models (LLMs) excel.

This strategy works only when combined with known assumptions about an image. Graph comprehension questions are embedded with strong priors such as standardized layouts (e.g., axes, legends) and may not require the model to reason over nonlocal regions. Models could instead learn to exploit these learned commonalities and only superficially parse graphs. This reliance on convention over direct visual evidence would stunt models from parsing images that defy these norms. Indeed, recent literature supports the hypothesis that VLMs are not as proficient at reading graphs as benchmark scores suggest [13]. For example, Masry et al. [13] showed that top models' performance catastrophically declined on their novel ChartQAPro dataset compared to ChartQA, indicating a missing skill set

for handling diverse, unseen charts. This, along with findings from benchmarks like HallusionBench [6], suggests VLMs still prioritize background knowledge over the visual evidence presented to them.

We identify three core types of nonlocal visual reasoning. First, *comparative perception* is the qualitative comparison of visual entities even when precise discrepancies are difficult to articulate (e.g., recognizing that two complex shapes are not identical without explicitly itemizing every differing feature). Second, *saccadic search*, named after the rapid eye movements humans make, is the process of gathering and integrating information from different image regions. Each piece of evidence informs the next step; for example, this process is used when consulting a chart's legend, then locating the corresponding line, then referencing an axis. Third, *smooth visual search* describes the continuous tracing of visual elements, such as following the outline of an object or tracing a curve to its conclusion. To systematically evaluate these capabilities, we introduce a procedurally-generated evaluation set comprising three task categories designed to be trivial for humans and require minimal prior knowledge. We present three task categories: *Object Re-Identification*, *Visual Scavenger Hunt*, and *Circuit Connections*. Examples of these tasks appear in Figure 1.

We design our evaluation to answer the following questions:

1. When do VLMs err in basic perception or perceive correctly but fail at visual reasoning? Do early perceptual mistakes compound or self-correct during reasoning?

2. Can VLMs perform *comparative perception* and *saccadic search*? If so, must these models use natural language judgments to guide these processes, or can they execute these tasks through direct visual analysis?

3. Can VLMs perform *smooth visual search*, an operation that involves tracing a continuous contour or path not easily decomposable into natural language steps? If VLMs find this continuous operation challenging, do they attempt to reframe it as a sequence of discrete operations, or use a different heuristic?

All tested models perform poorly on at least one variant of a task that is trivial for humans. The performance gradient on *Object Re-Identification* shows that modern models selectively choose when to examine an image closely. Our work further shows that fuzzy vision interferes with visual reasoning and the models seem unable to self-correct. They instead rely on their prior natural language judgments over direct evidence in the image. VLMs struggle the most with *smooth visual search* and our analysis of when they succeed suggests that most models cannot trace lines.

Our major contributions are:

- We release three procedural generators for an evaluation suite. Each generator creates synthetic image-question pairs for a minimal-context task to probe three distinct facets of nonlocal visual reasoning: comparative perception, saccadic search, and smooth visual search. Our generator, evaluation sets, and evaluation code are available here.

- We conduct a comprehensive evaluation of leading VLMs (including GPT-5, GEMINI 2.5 PRO, and CLAUDE SONNET 4) demonstrating that even flagship models lag far behind humans on trivial visual reasoning tasks, despite advances in primitive vision.

- We create several variants of each task to determine why models fail to perform as well as humans. We also determine under which circumstances they look carefully at images and use their perception abilities.

## 2   Related Work

**Benchmarks on Perception Primitives.** VLMs achieve strong performance on many complex tasks, including OCR, image captioning, and scene understanding [17, 2, 5, 22, 1, 15]. However, this high-level competence contrasts with known deficiencies in low-level perception. For instance, VLMs can struggle with recognizing basic shapes or performing simple visual arithmetic [16, 7, 20]. Research suggests these perceptual limitations may originate in the language decoder, even with adequate image encoder representations [7, 16]. These foundational gaps raise questions about whether VLMs' success always stems from robust visual processing, and critically, when and how they engage in visual algorithms.

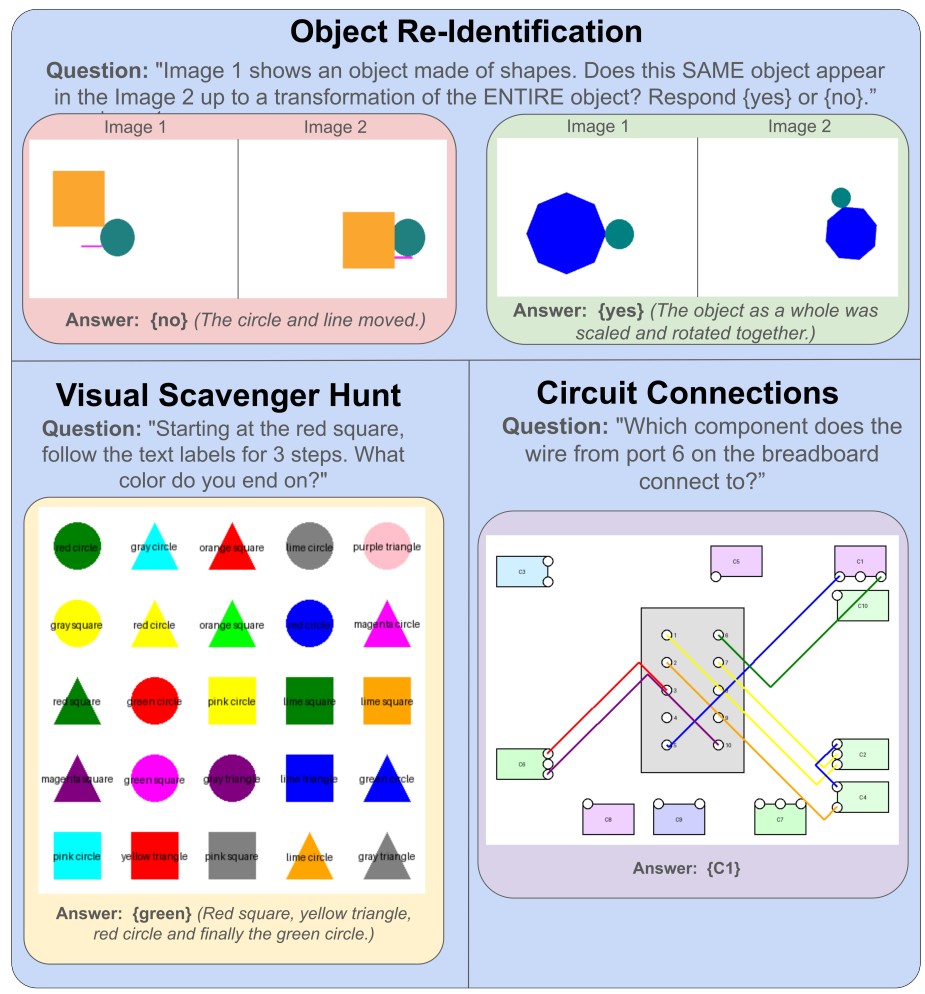

Figure 1: *Object Re-Identification* (top): Determine whether the same object that appears in Image 1 also appears in Image 2, up to a transform of the entire object but not individual component shapes. *Visual Scavenger Hunt* (bottom-left): From the indicated shape, follow the written labels for the specified count and report the final shape's color. *Circuit Connections* (bottom-right): From a named port on the central breadboard, trace the wire to its endpoint. The prompts here are abbreviated; full instructions are in the appendix.

**Benchmarks on Visual Reasoning.** To probe the underlying algorithmic visual skills necessary for robust image understanding, we take a different approach from existing visual reasoning benchmarks. Evaluations like Bongard problems and ARC [3, 4] test complex reasoning and are constructed as abstract puzzles that are challenging humans. Failing such a task does not distinguish between a failure in high-level abstract reasoning and a failure in fundamental visual processing. Our evaluation isolates these primitive visual skills by stripping away the puzzle-solving layer. While benchmark suites like VisOnlyQA and HallusionBench evaluate specific visual weaknesses, such as hallucination and illusion failures under controlled or adversarial settings [9, 6], we evaluate general visual algorithms.

**Chart and Graph Understanding.** The importance of visual data interpretation has motivated numerous evaluations and benchmarks such as ChartQA and MultiChartQA, which expose VLMs to diverse charts [12, 23, 8]. This area has also spurred the development of several models specifically trained for chart understanding [14, 11]. However, VLMs' underwhelming performance on more recent benchmarks such as ChartQAPro suggests that they have yet to develop robust graph-understanding capabilities [13].

# 3 Evaluation Design

Our evaluation isolates three core capabilities of nonlocal visual reasoning, each grounded in principles from classical human vision research. The first, *comparative perception*, draws on work in perceptual organization and attention. Classic Gestalt principles such as proximity and connectedness suggest that humans automatically group visual elements into coherent objects [21], while Treisman's Feature Integration Theory (FIT) [18] shows that attention serves to bind distributed features into a unified representation. By comparing *Object Re-Identification* performance on the Unconnected variant against the Standard variant (where components are contiguous), we directly test whether models are sensitive to the Gestalt principle of connectedness. The second capability, *saccadic search*, is motivated by Ullman's theory of visual routines [19], which proposed that complex visual operations are composed of primitive procedures such as shifting the focus of attention and indexing marked locations. Specifically, it is based on systematic scanning, or using each observation to guide the next. The third capability, *smooth visual search*, corresponds to the visual routine of boundary and contour tracing within Ullman's framework.

## 3.1 Object Re-Identification.

We use this task to test *comparative perception*: the model must hold two views in working memory and compare them under allowed transformations. Each instance of the task involves two images, 'Image 1' and 'Image 2'. 2–6 shapes are shown in 'Image 1'. In 'Image 2', this entire object undergoes a random transformation (i.e., rotation, translation, and scaling) and is always kept fully in-frame.

In half of the examples (the negative cases), one or more of the component shapes are also transformed independently to create a structurally different object. We avoid nearly imperceptible transformations, such as small rotations, translations, and re-scalings, as we do not aim to test pixel-level accuracy. Additionally, we render distractor objects in 'Image 2' to reflect the human capacity to re-identify objects across different environments. We restrict these distractor objects from occluding any components of the original object. Examples of this task can be found in Figure 1.

To disentangle VLM success conditions, we present three presentation variants. The Standard variant renders objects with physically contiguous component shapes to test the most intuitive concept of what an object is. The Unconnected variant removes this contiguity requirement to evaluate a more abstract object concept and determine how connectedness affects VLM perception. In the *Pixel-Perfect* variant, we do not apply any transformation to the object as a whole. Thus, positive examples of 'Image 2' are pixel-for-pixel matches of 'Image 1' (except for the added distractors). Examples of all three tasks can be found in the Appendix.

The examples are generated from a uniform distribution between "Yes" and "No", so the random guessing baseline is 50%.

## 3.2 Visual Scavenger Hunt.

This task tests *saccadic search*: the ability of the model to make discrete, evidence-driven jumps across the image. In many real-world visual challenges, locating a pixel cluster with semantic content is not enough. Instead, the task adapts as the agent gathers new clues. For example, an unlit gas stove does not signal a gas leak unless the knob is in the on position. Humans use each observation to decide where to look next, and this capability is vital to generalist agents.

We design our *Visual Scavenger Hunt* task to evaluate iterative visual search explicitly. The model is presented with a grid of different-colored shapes, each labeled with a different color-shape pair. The prompt provides a starting shape, and requires the model to follow a straightforward "scavenger hunt" of shapes around the image for a specified number of steps (which we call the chain length). The scene is randomly generated, so the entire scene must be searched to find a specific shape. A visual example can be found in Figure 1.

We evaluate the models on chain-lengths of 2, 3, and 4 to test if their performance degrades over long horizons. As we permute over 11 colors, the random chance baseline is approximately 9%.

### 3.3 Circuit Connections.

We frame *Circuit Connections* as an instance of *smooth visual search*. The model is shown a procedurally generated circuit diagram with a breadboard, several components, and wires. The task is to specify which component a breadboard port is connected to; the model must trace a continuous contour—a wire—from its source to its terminal point. An example of this appears in Figure 1.

Inspired by the "Subway Connection" task in BlindTest [16], we refine the task to directly evaluate the ability to follow the wire. Our synthetic generator contains three variants: the Standard, Single Color, and Unique Colors versions. On the Standard trial, each wire is one of five colors, selected randomly. The Unique Colors variant is designed as a control; each wire has a unique color so that the task is solvable by only looking at the ends of the wire. In the Single Color examples, all wires in an image have the same color. This trial is designed to prevent the model from taking shortcuts—by associating colors with locations or reasoning in natural language—rather than following the wire manually.

Although our previous tasks are motivated by abilities necessary for real-world tasks, we design *Circuit Connections* to mirror a practical skill: reading wiring diagrams. We adopt informal conventions—wires hold their directions at crossings—to improve diagram clarity and emphasize the need to trace the actual path. As we do not aim to test schematic-reading priors, we state these conventions explicitly in the prompt.

The rendered image contains between 4-10 components, chosen from a uniform distribution. Thus, a random guessing baseline is 14.29%.

## 4   Experiments

We evaluate GEMINI 2.5 PRO, CLAUDE SONNET 4, GPT-5, as well as other closed and open-source models on our benchmark in a few-shot setting. All error bars represent standard error. The full evaluation details, including prompts, can be found in the Appendix.

Before running our experiments, we manually self-evaluated 200 examples of each task's main category to find a human baseline. Our evaluators scored 100% on the *Object Re-Identification* and *Visual Scavenger Hunt* tasks and 99.5% on the *Circuit Connections* trial.

### 4.1   Object Re-Identification

The results for *Object Re-Identification* are summarized in Figure 2. On the Standard variant, no model significantly outperforms random chance, indicating that they cannot perform *comparative perception*. However, on the other two trials, models such as GPT-5, GEMINI 2.5 PRO, and CLAUDE SONNET 4 score between 12-24 percentage points higher. They remain over 20 percentage points below the human baseline in all trials.

The F1 (positive) and F1 (negative) scores, displayed in Figure 3, subdivide the models into three classes. Firstly, models that essentially ignore the input and almost always predict the same result (MOLMO 7B and LLAMA 3.2 11B). The vast majority of models attempt to answer the question and are not especially biased across all trials, but give poor predictions for both classes. The third class consists of models that improve in the latter two variants.

**Failure Modes**   A group of models—MOLMO (7B), LLAMA-3.2 VISION (11B), and PHI-4 MULTI-MODAL (14B)— effectively ignore the input and almost always predict the same result, as evidenced by their F1 scores being 0.00 for one class across multiple variants (Figure 3). This indicates they either cannot perform comparative perception or do not attempt to do so across all trials.

The models highlighted in the orange box in Figure 3 fail in ways that show they have the ability to selectively compare objects only in certain circumstances. On the Standard variant, their accuracy is near random chance, with the highest score being 60%. However, their performance significantly improves on both the Unconnected (GPT-5: 77%, GEMINI 2.5 PRO: 65%, CLAUDE SONNET 4: 79%) and Pixel-Perfect ( GPT-5: 71%, GEMINI 2.5 PRO: 65%, CLAUDE SONNET 4: 70%) variants. In the Standard variant, the component shapes of the object are always physically contiguous. According to Treisman's Feature Integration Theory, humans bind basic features—color, shape,

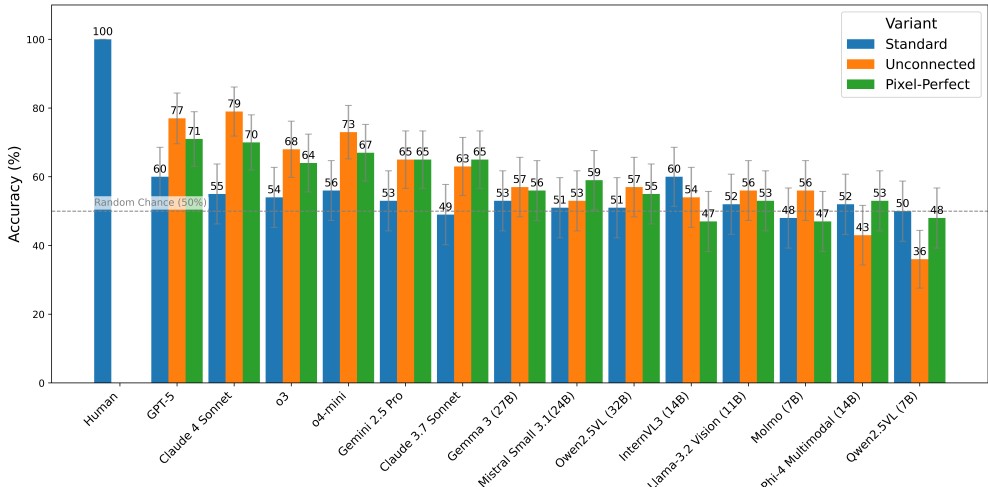

Figure 2: Accuracy on all variants of *Object Re-Identification*.

orientation—into unified object representations through focused attention. These models' lower accuracy on the Standard variant indicates they do not bind objects the same way humans do. More specifically, their poor performance (<60%) suggests that these models struggle with fine-grained inspection of connected entities. This contrasts with human perception, where Gestalt principles of grouping (e.g., by proximity and connectedness) make connected objects easier to process and compare. Because the Standard variant is a subset of the Unconnected variant, we suspect that these models are capable of comparing different objects but engage this ability selectively. If two Gestalt-grouped objects are similar enough the model does not inspect them closely. They appear to bypass comparative attention when objects appear similar enough under Gestalt grouping cues.

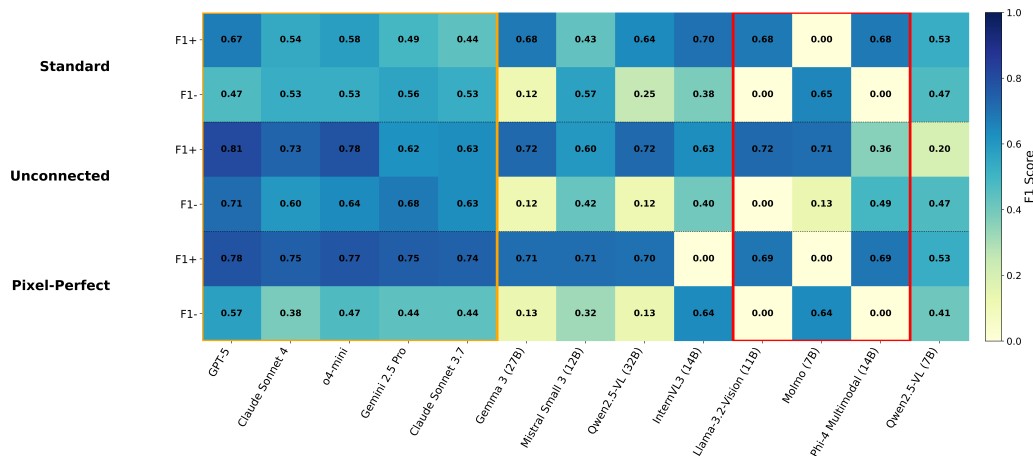

Figure 3: F1 for positive and negative classes across all trials of *Object Re-Identification*. Some models predict identically across the majority of trials (red box.) The strong models (orange box) perform poorly on the standard variant, but become better at recognizing similar objects when tested on the other two trials.

**Natural Language Strategies** The stronger models also perform slightly worse on the Pixel-Perfect as they do on Unconnected. On this trial, GEMINI-2.5-PRO scored 100% accuracy on the 11 responses in which it provided more than 10 tokens. We theorize that because the first and second images are so similar, these problems are easier to convert to an instance of natural-language comparison as *any* perturbation disqualifies the second object from being identical to the first.

Taken together, these results indicate that VLMs are currently incapable of generalist *comparative perception* and do not always examine the images carefully.

## 4.2 Visual Scavenger Hunt

The results for *Visual Scavenger Hunt* are shown in Figure 4. GEMINI 2.5 PRO, O4-MINI, and GPT-5 perform well above the random-guess baseline of 9%. O4-MINI shows a clear trend downward as chain length increases. For all models except GPT-5, O4-MINI, and GEMINI 2.5 PRO, performance is mostly static with respect to the length of the chain. CLAUDE 3.7 SONNET, PHI-4-MULTIMODAL, O3, and GEMMA 3 27B perform a few percentage points above this baseline. The rest of the models score at about the accuracy of random guessing.

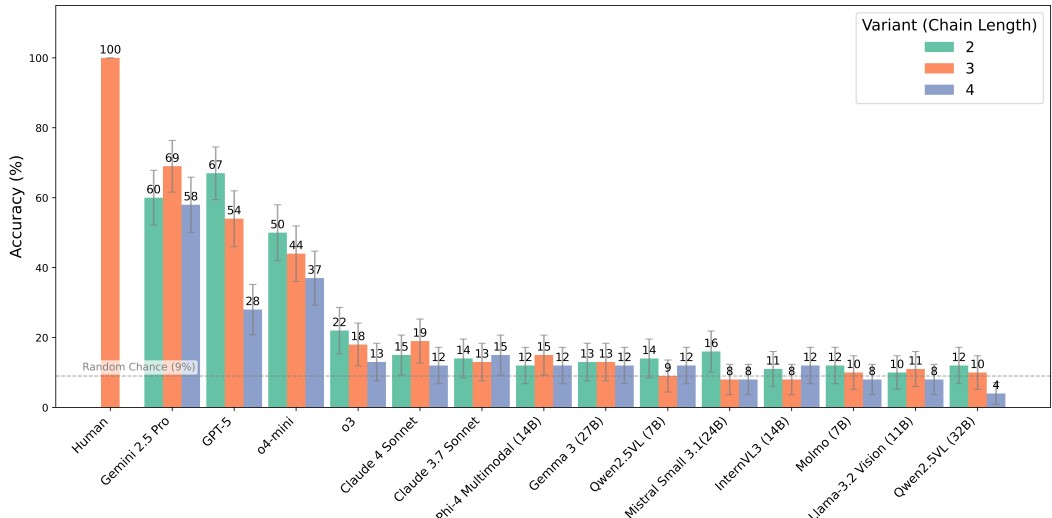

Figure 4: Accuracy on the *Visual Scavenger Hunt* task. Only GPT-5, O4-MINI, and GEMINI 2.5 PRO significantly outperform random chance.

**Failure Modes**   We manually observe the first 20 responses of all models. The weaker models exhibit guessing behavior. They hallucinate paths even for a chain length of 2; an example of this type of response can be found in Figure 5. Even with correct final colors, justifications often cite non-existent shapes or paths. O4-MINI often claims it cannot read the text and refuses to answer the question. We debunk this claim with an ad-hoc study; in isolation, O4-MINI was able to extract text out of a randomly selected tile on 20 random examples from our dataset. However, it cannot always accurately extract the entire grid, as shown in Figure 5. It successfully extracts every label, but hallucinates when it transcribes the actual shapes and colors. In our empirical analysis, all of GEMINI 2.5 PRO's mistakes occurred mid-search and seemed to be due to confusion with a nearby slot that had a similar color, label, or shape.

To determine if failures on this task stem from an inability to locate the shapes and text, or from a breakdown in chaining these steps through *saccadic search*, we performed a follow-up experiment. We decomposed the chain length 3 task into three sequential, single-step queries, where the model's answer for one step was used as the starting point for the next. The results are shown in Table 1. In addition to overall accuracy on the three-step sequence, we report the Final-Step Error Rate. This metric is calculated only on trials where the model correctly navigated the first two steps, thereby isolating the error rate of the final jump. We use this conditional metric because grading intermediate steps requires verifying both shape and color, whereas the final step, like our main task, only requires identifying the correct color.

These results show that top-performing models are highly capable of executing the atomic unit of the task when guided, whereas the other models cannot perceive well enough to perform the task. However, their high single-step accuracy contrasts with their low accuracy over a multi-step search. Simple error multiplication suggests O4-MINI should achieve 83% accuracy ($0.94^3$), far higher than

Table 1: Performance on the decomposed Visual Scavenger Hunt task over 75 trials. Accuracy reflects the success rate over the full three-step interactive sequence, while the Final-Step Error is conditioned on the first two steps being correct.

| Model | Accuracy (%) | Final-Step Error (%) |
|---|---|---|
| o4-mini | 90.67 | 6 |
| Gemini 2.5 | 88.00 | 8 |
| Llama 3.2 Vision (11B) | 24.00 | 85 |
| Qwen 2.5 VL (32B) | 16.00 | 86 |
| Qwen 2.5 VL (7B) | 14.67 | 85 |

the 44.5% observed in Figure 4. This discrepancy indicates that the primary bottleneck for the strong models is not perception, but the inability to perform multiple steps autonomously. Conversely, the other models' poor performance suggests they lack the basic perceptive abilities required to perform a single step correctly.

**Heuristic Strategies**   The marginal success of the second group of models over random accuracy likely results from simple heuristics, like choosing the most frequent color that appears in the image (a strategy with 12% accuracy), not task capability. This is evidenced by the lack of clear trend in their performance as chain length increases, which we would expect to decrease if they were performing a lossy visual algorithm.

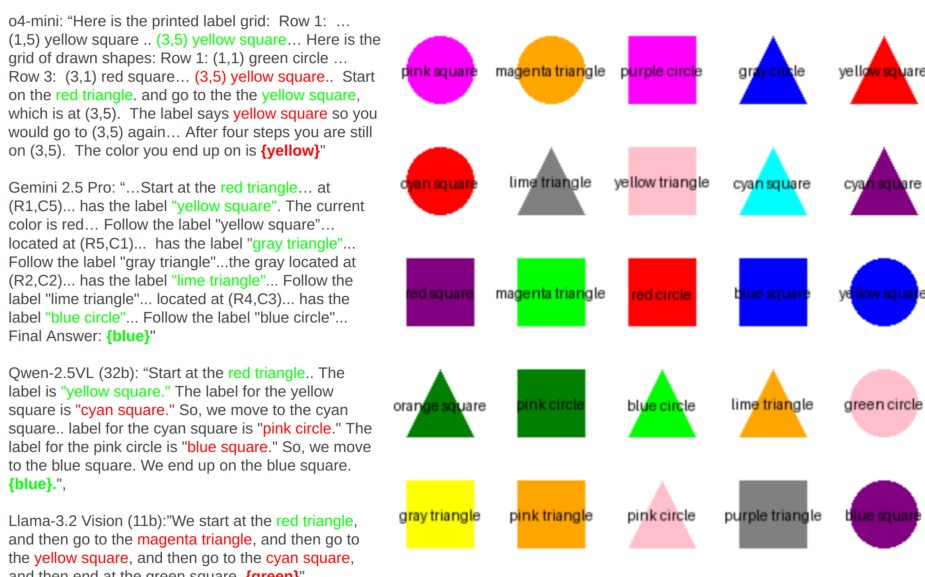

Figure 5: Example responses for *Visual Scavenger Hunt* from our qualitative analysis. Most models can locate the first shape but have trouble extending the chain from there. O4-MINI and GEMINI 2.5 PRO are both high-performing but use different strategies.

**Self Correction**   GEMINI 2.5 PRO and O4-MINI lack the ability to self-correct in this trial. Although examples with longer chains have more opportunities for error, they also provide additional signal that a mistake has been made. The other shapes in the image are labeled randomly, meaning about 30% of the time they lead to a shape that does not exist on the grid. For a human performing the task, this would be a strong signal of a prior mistake, but we do not observe this process across any model.

Most models lack the capability to systematically search through a novel image type that requires iterative visual grounding and evidence accumulation. Though the strongest models clearly outperform random accuracy, they make frequent mistakes, have inconsistent perception, and cannot self-correct.

## 4.3   Circuit Connections

Our results, summarized in Figure 6, indicate that all tested VLMs struggle with *smooth visual search*. Every model both exceeded random chance on the Standard variant and improved on the Unique Colors variant, with the exception of LLAMA-3.2. The hardest variant was Single Color, with a peak accuracy of 27% by GEMINI 2.5 PRO.

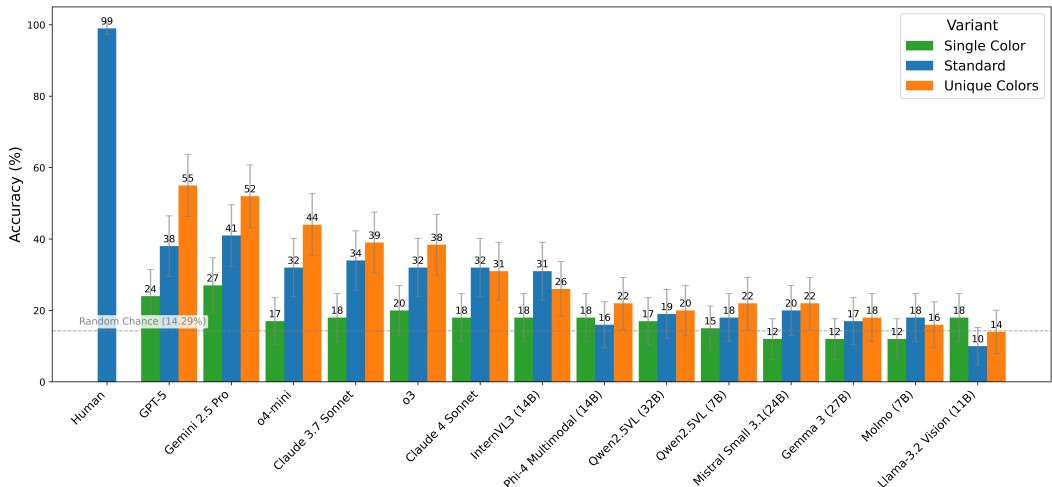

Figure 6: Results on the *Circuit Connections* task. Performance rises drastically across the board from Single Color to Unique Colors, suggesting that heuristics that do not involve tracing the line play a large role in model success.

Table 2 presents the log odds coefficients for two key characteristics: the Euclidean distance of the connecting wire and the number of times this wire crosses other wires. The log-odds analysis allows us to interpret by what factor the odds of a binary outcome (e.g., success vs. failure) change for each unit increase in a predictor variable, like the distance between the source and target port or the number of wire-crossings.

Table 2: $\Delta$ Log Odds and p-values for Distance (px) and Number of Crossings Effects for Circuit Connections. The log odds coefficient, $\beta$, for a characteristic indicates that a one-unit increase in that characteristic changes the logarithm of the odds of a correct model response by $\beta$. The null hypothesis is that the log-odds coefficient of 0.

| Model | Single Color Trial | | | | Standard Trial | | | | Unique Colors Trial | | | |
| | Distance Effect | | Crossings Effect | | Distance Effect | | Crossings Effect | | Distance Effect | | Crossings Effect | |
| | Log-odds | p-value | Log-odds | p-value | Log-odds | p-value | Log-odds | p-value | Log-odds | p-value | Log-odds | p-value |
|---|---|---|---|---|---|---|---|---|---|---|---|---|
| GPT-5 | -0.0106 | 0.00131 | -1.0008 | 0.00238 | -0.0063 | 0.00225 | +0.0028 | 0.982 | -0.0026 | 0.299 | -0.3477 | 0.0742 |
| Gemini 2.5 Pro | -0.0084 | 0.00546 | -0.5349 | 0.0244 | -0.0086 | $6.45 \times 10^{-5}$ | -0.1656 | 0.204 | -0.0055 | 0.0333 | -0.2761 | 0.151 |
| o4-mini | -0.0186 | $7.81 \times 10^{-5}$ | -0.7154 | 0.0312 | -0.0102 | $1.1 \times 10^{-5}$ | -0.3925 | 0.0142 | -0.0033 | 0.189 | -0.0519 | 0.782 |
| Claude Sonnet 3.7 | -0.0106 | 0.00328 | -0.9305 | 0.0102 | -0.0092 | $4.62 \times 10^{-5}$ | -0.2021 | 0.153 | -0.0097 | $6.94 \times 10^{-4}$ | -0.1130 | 0.561 |
| Claude Sonnet 4 | -0.0134 | $6.94 \times 10^{-4}$ | -0.5653 | 0.0561 | -0.0123 | $7.65 \times 10^{-7}$ | -0.0420 | 0.749 | -0.0097 | 0.00106 | -0.7260 | 0.00853 |
| InternVL3 (14B) | -0.0053 | 0.114 | -0.0587 | 0.798 | -0.0062 | 0.00363 | +0.0727 | 0.567 | -0.0121 | $1.95 \times 10^{-4}$ | -0.3154 | 0.191 |
| Phi-4 Multimodal (14B) | $+8 \times 10^{-4}$ | 0.791 | $-4 \times 10^{-6}$ | 0.999 | -0.0070 | 0.0082 | -0.0935 | 0.595 | -0.0015 | 0.594 | +0.0193 | 0.931 |
| Qwen2.5VL (32B) | -0.0053 | 0.114 | -0.0587 | 0.798 | -0.0137 | $2.63 \times 10^{-6}$ | -0.1847 | 0.291 | -0.0080 | 0.0129 | -0.8284 | 0.0209 |
| Qwen2.5VL (7B) | -0.0065 | 0.0692 | -0.2718 | 0.308 | -0.0042 | 0.0839 | +0.0236 | 0.878 | -0.0083 | 0.0084 | -0.2162 | 0.386 |
| Mistral Small 3.1 (24B) | -0.0160 | 0.00103 | -0.2182 | 0.449 | -0.0108 | $4.53 \times 10^{-5}$ | -0.2331 | 0.188 | -0.0081 | 0.00918 | -0.2546 | 0.309 |
| Gemma 3 (27B) | -0.0133 | 0.00321 | -0.5133 | 0.136 | -0.0165 | $5.01 \times 10^{-7}$ | -0.1152 | 0.509 | -0.0121 | $8.13 \times 10^{-4}$ | -0.4617 | 0.139 |
| Molmo (7B) | +0.0035 | 0.343 | -0.4027 | 0.21 | -0.0046 | 0.107 | +0.2088 | 0.188 | -0.0030 | 0.368 | +0.0790 | 0.746 |
| Llama-3.2 Vision (11B) | -0.0020 | 0.521 | -0.4820 | 0.0868 | -0.0141 | $6.07 \times 10^{-5}$ | -0.2813 | 0.252 | -0.0079 | 0.0303 | -0.1177 | 0.683 |

The statistics in Table 2 also suggest that these models are not performing human-like tracing. The strongest predictor of model performance across all trials is distance, with high confidence ($p \leq 0.05$) across almost all models on the Standard and Unique Colors trials. The crossings effect is strongest in the highest-performing models.

**Failure Modes**   Across all Circuit Connections trials, the models' success rates and our log odds analysis indicate that no model performs contour tracing. They cluster into two groups— one which does slightly better than random on Unique Colors and Standard, and the three models which significantly beat random accuracy.

Most models we tested show a complete inability to perform line tracing. These models perform at or below random chance (14.29%) across all three variants of the Circuit Connections task (Standard, Single Color, Unique Colors), as depicted in Figure 6.

This widespread low accuracy suggests these models cannot trace lines, and our log-odds analysis in Table 2 supports this assertion. Many models in this group show a significant correlation between Euclidean Distance and success in the Standard or Unique Colors trials when they beat random accuracy. However, the statistical significance of this effect disappears in the Single Color variant (e.g., MOLMO (7B) p=0.343) The wire-crossings effect for this group remains largely insignificant across all variants. This general lack of sensitivity to crossings is evidence they do not perform human-like line tracing— intuitively, the task of line-tracing is hardest when two lines interfere. We hypothesize they are able to exceed random accuracy by using color-cues and proximity heuristics.

Top performing models, such as GEMINI 2.5 PRO, CLAUDE 3.7 SONNET, and O4-MINI, form a second group that performs better but still demonstrates significant limitations in tracing. Their accuracy drops significantly when color cues are removed or made uniform. For example, GEMINI 2.5 PRO's accuracy falls from 48% on Unique Colors to 27% on Single Color. While tracing is expected to be harder when wires of the same color interfere, this low performance suggests either a lack of, or severely limited, tracing ability. The Euclidean distance of the connecting wire remains a significant negative predictor for these models across all trials. However, what distinguishes them from the first group is that these models do exhibit a statistically significant negative crossing effect in the challenging Single Color trial, where wires often cross other wires of the same color (e.g CLAUDE 3.7 SONNET shows a log-odds coefficient of -0.9305 (p=0.0102) for the crossings effect in this variant.) This behavior is consistent with a tracing algorithm, where crossings represent visually ambiguous points. However, because these models also do poorly in Unique Colors compared to humans, we suspect these models are possibly performing choppy, saccadic movements to find successive points on the wire rather than a smooth trace. This strategy aligns with the statistics: in the Single Color variant, models get caught in wire crossings and struggle to accurately jump to the next segment, impacting performance.

## 5    Conclusion

We introduce a suite of targeted tasks to distill *nonlocal visual reasoning* into three components: *comparative perception*, *saccadic search*, and *smooth visual search*. Most VLMs tested fail each benchmark, even those designed to deal with structured data. Closed-source VLMs achieved the highest accuracy but performed wildly differently under slight permutations of the underlying task. The task with the lowest performance ceiling was Circuit Connections, which requires *smooth visual search*. This suggests a principal limitation of these models is their difficulty with tasks that resist natural language reformulation.

This suggests that despite gains on other visual benchmarks, VLMs lack a reliable framework to analyze images. Without these capabilities, the vision of VLMs will remain fundamentally less robust than human vision.

**Limitations.**    While our synthetic setting isolates primitives, it does not capture the complete gamut of natural images, which may only require visual skims to parse correctly. Additionally, we evaluate on only 200 or 125 examples per variant for cost-efficiency reasons.

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

# A   Task Details: Examples and Prompts

This section provides visual examples for each task category, its variants, and the prompts used.

## A.1   Object Re-Identification

In this task the model must determine if an object in 'Image 1' is identical to an object in 'Image 2' under allowed transformations, possibly with distractor shapes present. For clarity, we provide examples for its three variants.

Figure 7 shows examples for the *Standard* variant where object components are contiguous. Figure 8 illustrates the *Unconnected* variant where object components are not necessarily connected. Figure 9 displays the *Pixel-Perfect* variant, where positive examples have no rigid transformation applied to the object between 'Image 1' and 'Image 2' (though distractors may be present in Image 2).

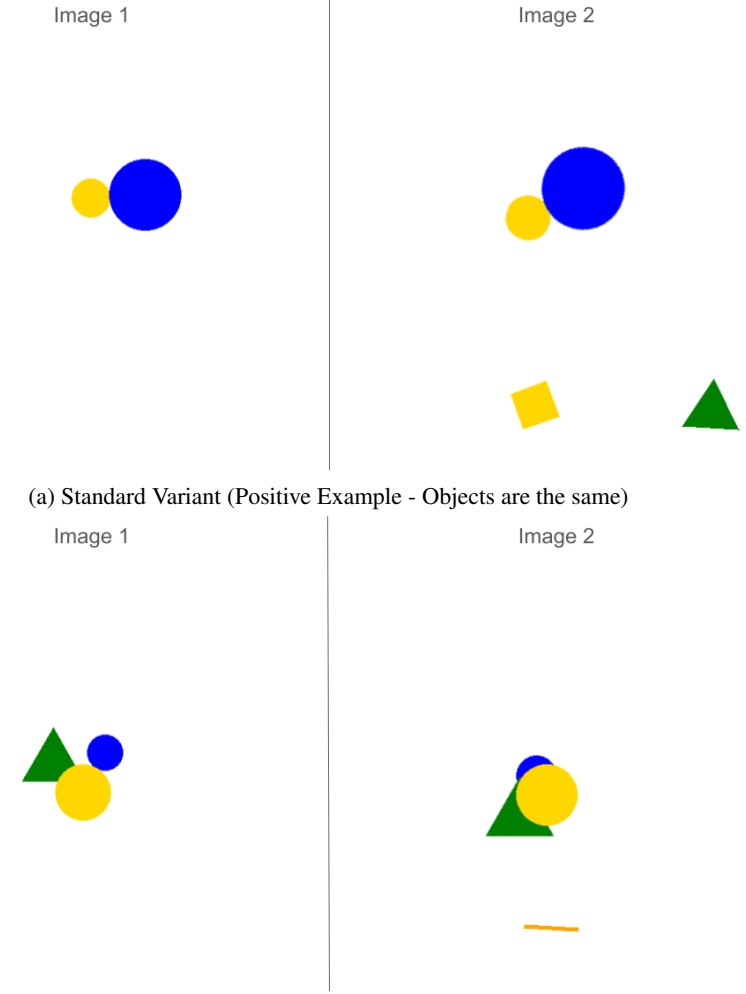

(a) Standard Variant (Positive Example - Objects are the same)

(b) Standard Variant (Negative Example - Objects are different)

Figure 7: Examples of the *Object Re-Identification (Standard Variant)* task. Prompt: "The first image shows an object made of connected geometric shapes, which together form an object. Does this SAME object appear in the second image? For example, if a component shape were to be rotated or translated separately from the entire composite-object, it would be a different object. Respond with {yes} or {no} (inside the curly brackets). There may be extra shapes in Image 2 that are not part of the original object; as long as the object from Image 1 is present, the answer is yes even if there are other shapes present." [Two examples with answers would precede this.]

## A.2 Visual Scavenger Hunt

In *Visual Scavenger Hunt*, the model is presented with a grid of different-colored shapes, each labeled with a different color-shape pair, as shown in Figure 10. The image generation procedure for the grid is identical across variants; the only change is the chain length (number of steps) the model must follow.

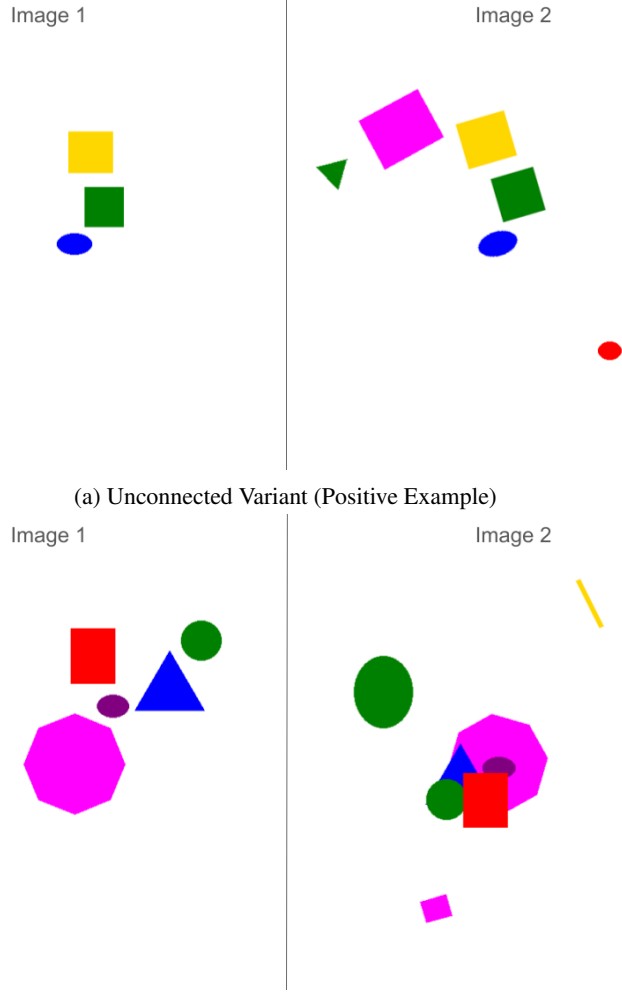

(a) Unconnected Variant (Positive Example)

(b) Unconnected Variant (Negative Example)

Figure 8: Examples of the *Object Re-Identification (Unconnected Variant)* task. Prompt: "The first image shows an object made of geometric shapes, which together form an object. Does this SAME object appear in the second image up to a rigid translation, rotation, and scale of the ENTIRE object as a whole? Respond with {yes} or {no}. There may be extra shapes in Image 2 that are not part of the original object; as long as the object from Image 1 is present, the answer is yes even if there are other shapes present." [Two examples with answers would precede this.]

### A.3 Circuit Connections

The *Circuit Connections* task (Section 3.3) assesses *smooth visual search*, requiring the model to trace a wire from a specified port on a central breadboard to its connected component. Examples for its three variants, based on wire coloring, are shown in Figure 11: the Standard Variant (Figure 11a), where multiple wires can share colors; the Single Color Variant (Figure 11b), where all wires are the same color; and the Unique Colors Variant (Figure 11c), where each wire has a distinct color.

## B Evaluation Methodology and Supplementary Information

### B.1 Evaluation Parameters

Certain models—specifically MOLMO (7B) and LLAMA 3.2 VISION (11B)—technically accept multiple images as input, although their documentation advises against this practice. Accordingly,

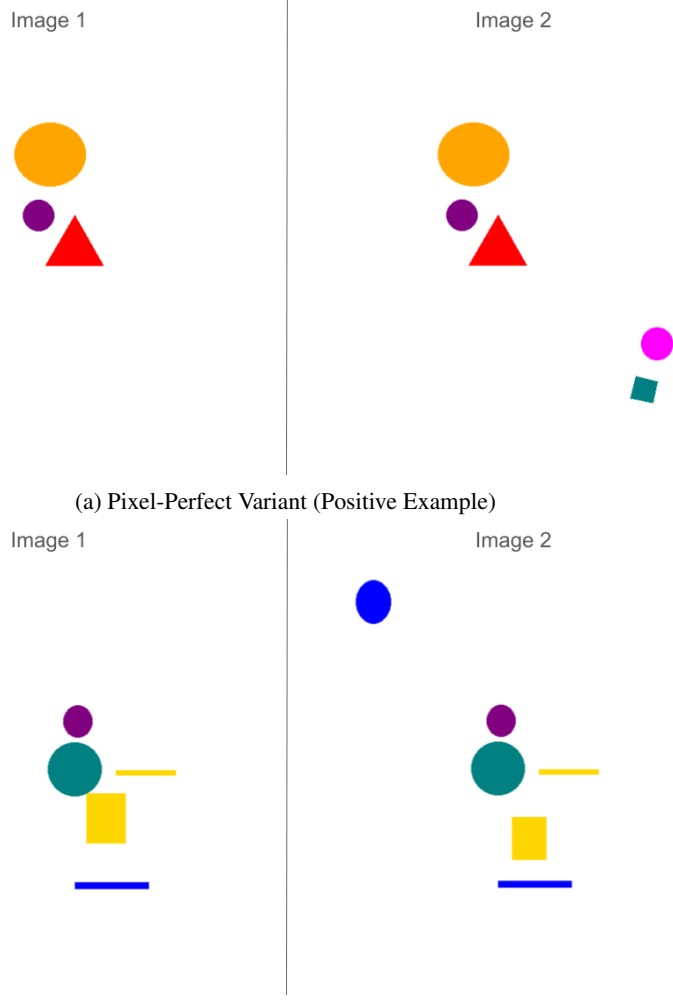

(a) Pixel-Perfect Variant (Positive Example)

(b) Pixel-Perfect Variant (Negative Example)

Figure 9: Examples of the *Object Re-Identification (Pixel-Perfect Variant)* task. Prompt:"The first image shows an object made of geometric shapes, which together form an object. Does this SAME object appear in the second image? For example, if a component shape were to be rotated or translated, it would be a different object. Respond with {yes} or {no} (inside the curly brackets). There may be extra shapes in Image 2 that are not part of the original object; as long as the object from Image 1 is present, the answer is yes even if there are other shapes present." [Two examples with answers would precede this.]

we ran each of these models twice in the few-shot setting: once with the image-based demonstrations and once without. We report the higher score of the two runs in every case. For LLAMA 3.2 VISION (11B) the inclusion of additional example images yielded higher accuracy on every run. MOLMO (7B) attained the best performance when the demonstrations were omitted.

Secondary metrics for o3 were captured incorrectly due to a bug in the evaluation code. They are thus omitted from Figure 3 and Table 2.

### B.1.1 Computational Resources

Experiments were conducted using a combination of local high-performance computing resources and commercially available model APIs. For the open-source models, we evaluated locally on a system equipped with three NVIDIA A6000 GPUs (48GB VRAM each) and dual Intel(R) Xeon(R) Gold 5220R CPUs @ 2.20GHz. For other models, we accessed them via API. We

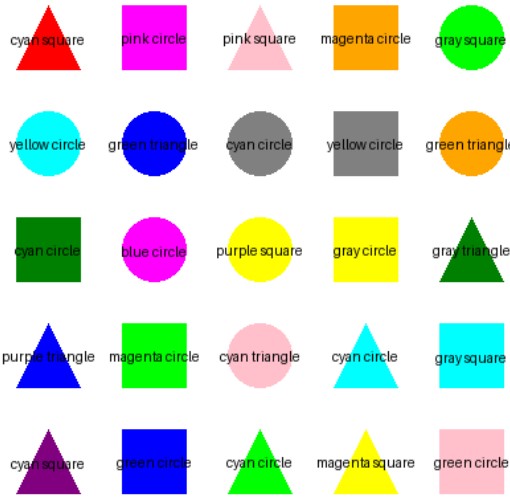

Figure 10: Example of the *Visual Scavenger Hunt* task. For a chain length of 4, the prompt would be: "Starting at the blue triangle, follow the labels for 4 steps. (For instance, in a different example of 4 steps, you might start at a blue triangle, then go to a red square, then a blue circle, then a magenta triangle, then a green circle. The answer would be green.) After those steps, what color are you on? Answer with the color in curly braces, e.g. {red}." [A single example tracing a path (accompanied by an image) would precede this, e.g: "We start at the blue square, and then go to the purple triangle, and then go to the yellow square, and then go to the red square, and then end at the gray square. {gray}" ]

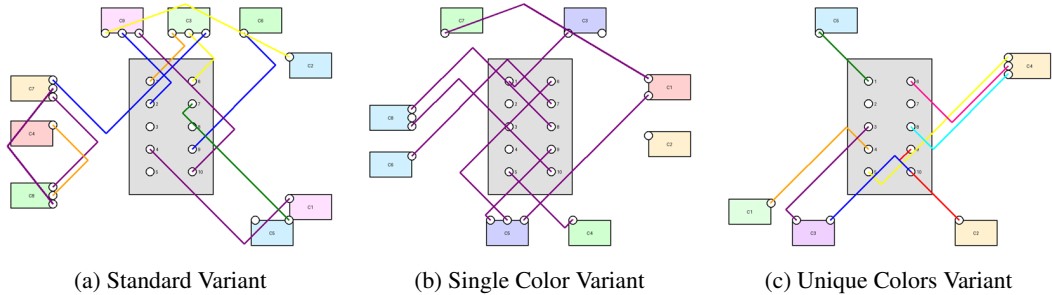

| (a) Standard Variant | (b) Single Color Variant | (c) Unique Colors Variant |

Figure 11: Examples of the *Circuit Connections* task variants. The prompt: "Which component does the wire from port 7 on the breadboard, which is the gray rectangle with numbered ports, connect to? A wire is a series of connected, same colored lines that go from the center of a port, represented on the screen as a white circle, to another port. Each wire only connects two ports, one at either end. A wire will NEVER turn at the same spot that it intersects another wire, and wires do not change colors. Answer with the component label in curly braces, e.g {C0}." [Two examples would precede this.]

evaluated CLAUDE 3.7 SONNET, GEMINI 2.5 PRO, and CLAUDE SONNET 4 via OpenRouter at model codes `anthropic/claude-3.7-sonnet:thinking`, `google/gemini-2.5-pro-preview`, and `anthropic/claude-sonnet-4`, respectively. These OpenRouter models were evaluated between May 1, 2025 and May 15, 2025 (for CLAUDE 3.7 SONNET and GEMINI 2.5 PRO) and between September 17, 2025 and September 20, 2025 (for CLAUDE SONNET 4). The OpenAI models were evaluated via the OpenAI API at model codes `o4-mini-2025-04-16`, `o3-2025-04-16`, and `gpt-5-2025-08-07`. We gathered model responses for these evaluations between April 28, 2025 and May 14, 2025 for `o4-mini-2025-04-16`, between July 24, 2025 and July 27, 2025 for `o3-2025-04-16`, and between September 17, 2025 and September 20, 2025 for `gpt-5-2025-08-07`.

