# OpenReview forum: "VLMs have Tunnel Vision: Evaluating Nonlocal Visual Reasoning in Leading VLMs"
_NeurIPS.cc/2025/Conference — NeurIPS 2025 spotlight_

### Official Review · Reviewer_2sTq · 2025-06-08

**Clarity:** 3
**Significance:** 2
**Originality:** 3
**Rating:** 4
**Confidence:** 3

**Summary:**

This paper presents a new benchmark evaluating nonlocal visual reasoning in VLMs. It is composed of three different tasks designed to test different types of nonlocal reasoning (comparative perception, saccadic search, smooth visual search). Results show that SOTA VLMs mostly fail to perform these tasks, which humans can easily complete, suggesting ample room for improvement in their reasoning capabilities.

**Questions:**

Would SOTA VLMs also mostly perform randomly on these reasoning types when they are required to understand natural images rather than synthetic shapes?

Are the reasoning types (e.g. comparative perception) inspired by studies in psychology? If so, it would be nice to provide references to their use in that field.

Does the object re-identification task connect to existing work on Image Difference Captioning or other methods to compare two images?

L174 How is the Circuit Connections task different from the existing Subway Connections task?

**Ethical Concerns:**

["NO or VERY MINOR ethics concerns only"]

**Final Justification:**

My main concerns were well-addressed in the author response.

**Limitations:**

Yes.

**Quality:**

2

**Strengths And Weaknesses:**

*Strengths:*

As VLMs improve, benchmarking their abstract reasoning abilities is becoming more important.
It is interesting to see tasks that are inspired by human psychology, which can be a fruitful avenue for finding new ways to test these models.

The idea of testing VLMs for sequential visual reasoning, particularly decomposed into three different sub-categories, appears to be novel and it is interesting to what degree current VLMs encapsulate them.

The results on the visual scavenger hunt task seem interesting, since there is a clear differentiation between models that do and do not show non-trivial performance.

*Weaknesses:*

I have some major concerns about the methodology which make me question the degree to which the results in the paper are meaningful.
My central concern is that these images depicting abstract shapes are very different from the natural images that these models are normally trained and evaluated on. This has already been explored in prior work (L22). If other works already show that modern VLMs don’t understand basic geometry in abstract shapes, what does this work add? In other words, do these results really say something interesting about different types of reasoning, or are they just another demonstration of this known phenomenon?

I also have concerns with the task images and prompts used (Fig 1). Some of the textual prompts use ambiguous and confusing wording. For example, the first task uses the term “object” in an abstract sense that is not obvious, and the second example in the figure states that the correct answer is YES because the object was scaled and rotated when scaling might not be considered a rigid transformation. In the third task, a phrase like “starting at the teal triangle” might be interpreted either to mean the given geometric shape and color, or the text stating “teal triangle”. It could be that VLMs struggle because they are not as good at interpreting confusing or convoluted prompts while humans manage to do so – this doesn’t necessarily measure higher level reasoning.

Similarly, the text in the image prompts is small and hard to read. Although you partially tested this in Supp A.2 (on only one of the many texts in images for one of two tasks using visual text), it could still be that results would improve if text resolution were increased.

Some less significant issues are:
* Some of the presentation can be improved. I found Fig 1 and its caption to be a bit convoluted, and could benefit from numbering the tasks or even from splitting into three figures accompanying each task in the text.
* L279 seems like an overclaim. Even if poor results are because of a gap in reasoning abilities, it is not clear that explicit structured reasoning is the solution. For example, maybe inference compute scaling methods without explicit structure such as chain-of-thought could help.

---

> ### Author Rebuttal · Authors · 2025-07-31
>
> We appreciate your detailed feedback, and respond below.
>
> ## Images depicting abstract shapes are very different from the natural images that these models are normally trained and evaluated on
>
> These shapes are not different from the training data of these models because they have been trained on charts, and diagrams, which make liberal usage of abstract shapes (See technical reports for Qwen, Phi, etc). We evaluate in this setting because it is an effective way of isolating visual skills from learning natural correlations (e.g a chair is usually near a table).
>
> ## If other work shows that VLMs don’t understand basic geometry, what does this work add?
>
> We test basic visual reasoning which no other benchmark tests. Previous work tested VLMs’ capacity to perceive abstract shapes and showed that older models could not count shapes or tell if they intersected. However, SOTA models (e.g., o4-mini) now achieve 90% accuracy on "VLMs Are Blind", which shows they do understand basic geometry.
>
> These models perform at around random accuracy on some portions of our benchmark, and significantly below humans on all portions. This discrepancy means that our tasks assess something more demanding: the ability to integrate multiple, nonlocal visual cues. Our evaluation allows us to prove that even though models can match humans on higher level tasks such as chart understanding, they cannot use the skills that humans use to solve those tasks (L33-39)
>
>
> ## Object Re-ID uses the term “object” in an abstract sense that is not obvious
>
> The full prompt specifies an object is made of geometric shapes; the reviewer is referring to the prompt in Fig. 1, but this prompt was truncated to simplify the illustration. In Fig. 1,2 and 3 in the supplement, we show the full prompt:
> “The first image shows an object made of connected geometric shapes, which together form an object. Does this SAME object appear in the second image?...”
>
>
>
> ## The 2nd example in Fig. 1 states that the correct answer is YES because the object was scaled and rotated when scaling might not be considered a rigid transformation.
>
> The full prompt (supplement, Figure 2) specifies that scaling counts as a transformation. Additionally, only the Unconnected variant uses the term “rigid transformation". The full sentence is as follows: “Does this SAME object appear in the second image up to a rigid translation, rotation, and scale of the ENTIRE object as a whole?”
>
> ## In the third task, a phrase like “starting at the teal triangle” might be interpreted either to mean the given geometric shape and color, or the text stating “teal triangle”.
>
> This prompt is unambiguous because it is evaluated with a solved example in its context. On its own, this phrasing is ambiguous, but we also provided the model with a different solved example of VSH, fully annotated with the correct chain (“We start at the red triangle, then go to the blue triangle…”). This chain only matches the image if you start with the shape and then use its text to keep going.
>
> We ran an additional experiment on VSH that shows that the high performing models understand the prompt but cannot perform saccadic search. We modified the chain length 3 trial to go one jump at a time per model response. We used the model’s answer for the prior step as the starting point for the next request.
>
> Gemini 2.5: 88%
>
> o4-mini: 90.6%
>
> Qwen 2.5 7B: 14.7%
>
> This much-higher success rate from SOTA VLMs indicates they know to start on the shape.
>
> This experiment also allows us to capture the per-step error rate of each model. Gemini-2.5 has a final-step failure rate of just 8%, and o4-mini’s is just 6%. Yet, assuming independent errors and using Fig. 4’s overall accuracies (68.5% and 44.5%), we’d predict per-step error rates of 12% (1−0.685¹ᐟ³) and 24%. The discrepancy shows that saccadic search, not individual step comprehension, drives the performance drop.
>
>
>
> ## Maybe VLMs struggle because they are not good at interpreting confusing prompts
>
> Our experiment in the teal triangle response above and prior research suggest that it is unlikely the VLMs struggle with our tasks because they don’t understand the prompts. Research about LLM’s instruction following ability is mixed, and some studies show instruction tuned models exceed human performance on ambiguous instructions (AmbiBench, Are Language Models Worse than Humans at Following Prompts? It's Complicated).
>
> Additionally, understanding the few-shot examples also requires the skills that we test. (L191) In the reviewer’s specific example—whether to start with the teal triangle’s text or its shape—the model could resolve that ambiguity by performing a saccadic search over the solved example in its context.
>
> Our experiment in the section above further strengthens our claim that the models understand the individual steps but cannot search. The models are more accurate when given the same prompt but for a single step— it is search that is difficult.
>
> ## The text in the image prompts is small and hard to read, and this could hurt performance
> We run new experiments and show that text size does not lower performance on Circuit Connections(CC) and does lower performance on Visual Scavenger Hunt(VSH). However, our experiments show that models struggle more with saccadic search than text-reading.
>
> We generated an additional 75 examples of the Standard Variant of CC and tripled the text size. Here are the results:
>
> Gemini 2.5 Pro: 32%
>
> Claude 3.7 Sonnet: 32%
>
> Gemma 27B: 17.33%
>
> Qwen 7b: 21.33%
>
> No model performs significantly better than in Fig. 6, suggesting that text size does not hurt performance.
>
> We also conducted two experiments for VSH, Chain Size 3. In the first one, we doubled the text size. In the second, we replaced the text labels with mini-shapes and modified the prompt to be as follows:
>
>  "This is a grid of LARGE shapes (outlined in black), each containing a small mini shape (outlined in white) inside of it. You need to follow a chain by looking at the mini shapes inside the large shapes… For example: If you start at a large red square that contains a small blue circle inside… The answer would be {yellow}. Always move to the LARGE shape that matches the mini shape you just saw. Starting from the large purple triangle, follow the chain for 3 steps… Answer with the color in curly braces, e.g. {red}."
>
> The results for the large-text trial are as follows:
>
> Gemma 3 27B: 17.4%
>
> Qwen 7b: 9.33%
>
> Gemini 2.5 Pro- 77.3%
>
> o4-mini-high: 81.33%
>
> And for the no-text version, they are as follows:
> Qwen 7b: 18.67%
>
> Gemma 3 27b: 5.67%
>
> Claude 3.7: 29.33%
>
> Gemini-2.5 Pro: 56%
>
> As Gemini 2.5 Pro was 68.5% accurate and o4-mini was 44.5% accurate in the original trial (Fig. 4),  high‑performing models perform worse with smaller text size on VSH. In the Unclear Prompts analysis, o4‑mini achieves ~94% one‑step accuracy—so, under independent‐error assumptions (0.94³≈83%), three‑step performance should be ≈83%, but Fig. 4 shows only 44.5%. This shows that they struggle more with saccadic search than text reading.
>
> ## “I found Fig. 1 and its caption to be a bit convoluted”
>
> We will reorganize Fig. 1 to make the individual tasks clearer.
> ## “These findings strongly argue for a shift in focus towards models that explicitly support structured and systematic visual reasoning...”(L279) is an overclaim. It is not clear that explicit structured reasoning is the solution. Maybe inference compute scaling methods could help
>
> We will tone it down. We did not mean that explicit structured reasoning will necessarily be the solution. Other approaches could help too. We will revise accordingly.
>
>
> ## Do SOTA VLMs perform badly on these reasoning tasks over natural images?
>
> We performed new experiments that show that these models also perform poorly on semi-natural images, which suggest that models may also struggle on fully natural images.  We modified VSH to use natural images from the COCO dataset and render labels on the objects which refer to other objects in the scene. We use image segmentation data to find the pixels that belong unambiguously to one object and render text labels at those locations.  We evaluate several models on this in a two-shot setting.
>
> Claude 3.7: 33%
>
> Gemini 2.5: 38.67%
>
> Qwen 2.5 7b: 26.67%
>
> Qwen 2.5 32b: 40.00%
>
> Gemma 3 27b: 40%
>
> On COCO images with ∼4 labels per image, the VLMs achieve only 26–40% accuracy, barely above the ~25% they would get by randomly picking a label. These results show that the failure persists even in semi-natural images.
>
> ## Are the reasoning types inspired by psychology?
>
> Yes, our reasoning types are loosely inspired by psychological research, and we will update  the paper to include references. We now cite Ullman’s (1984) visual routines framework—whose “scan” and “trace” routines mirror our saccadic search task—and Treisman’s (1980) Feature Integration Theory to explain VLMs’ failures in object‐feature comparison (L209–214).
>
> ## Does object re-id connect to existing work in Image Difference Captioning(IDC)?
>
> Object Re-ID evaluates a sub-task of IDC. Additionally, while IDC aims to describe discrepancies, evaluating if a description is correct is difficult. Evaluating binary correctness instead of a caption makes our evaluation precise.
>
>
> ## What is the difference between Circuit Connections and BlindTest’s Subway Connections?
>
> Subway Connections can be solved without smooth search, whereas our benchmark requires smooth search and so can be used to test it. The Subway Connections task can be solved by looking at A and B, seeing which colors leave each of them, and the number of connections is just the intersection between those two. In our Single Colors task, this is impossible because all wires are the same color, so the only consistent way to solve it is to actually trace the line. Only our Unique‐Colors control does the task resemble Subway Connections, and we use this as a control to examine which factors cause success and failure in each setup.

---

> > ### Comment · Reviewer_2sTq · 2025-08-01
> >
> > Thank you. This addresses my main concerns and I will update my score accordingly.

---

### Official Review · Reviewer_3qvb · 2025-06-16

**Clarity:** 3
**Significance:** 2
**Originality:** 3
**Rating:** 4
**Confidence:** 3

**Summary:**

This paper is an empirical study on sequential and non-local spatial reasoning in VLMs. The paper introduces an evaluation suite for 3 such reasoning tasks and demonstrates that current VLMs generally fail at these tasks even though they are mostly trivial for humans to complete.

**Questions:**

1. Fig 2, 4, 6 do not have the variants for o3. Is there a reason these results are missing?
2. The discussion in Section 4.2 around the justifications is insightful, as it shows that these tasks can fundamentally be broken down into smaller subtasks which the model should be able to solve. This suggests potential strategies for mitigation. How do the models perform in the chain length = 1 variant? Can you extensively verify which tasks are feasible for breaking into small parts and solving and which are not?
3. Similarly, the viability of self-correction is an interesting consideration. Is the model aware of the viability of a self-correction strategy, e.g., is it aware in the prompt that some texts point to non-existent objects?
4. In Sec 4.3, it is interesting to identify that the model is not performing tracing. Can we hypothesize what approach the model is taking to solve the problem?

**Ethical Concerns:**

["NO or VERY MINOR ethics concerns only"]

**Final Justification:**

Most of my questions were answered and resolved. My remaining question is how to overcome poor performance. The authors suggest some ideas around breaking down tasks or separately executing visual steps. I look forward to seeing some of these ideas play out in future work

**Limitations:**

Yes

**Quality:**

3

**Strengths And Weaknesses:**

Strengths:
The evaluation suite is of great value to the community as well as the main findings around current limitations of VLMs. The empirical analysis is also thorough across many different VLMs to definitively demonstrate the key finding.

Weaknesses:
The paper primarily focuses on identifying an existing weakness within VLMs. While this work is useful and important, the analysis towards identifying the root causes and limitations of the weakness is limited. Finally, there is limited discussion towards overcoming the weakness. Note that while the paper is not required to present a complete positive result in fully resolving the issue, it can be strengthened with some reasonable direction towards improvement.

---

> ### Author Rebuttal · Authors · 2025-07-31
>
> We thank the reviewer for their feedback. We are encouraged that they find our evaluation suite thorough and valuable, and have addressed their concerns below.
>
> ## The analysis of identifying the root causes of weakness and ways to overcome the poor performance is limited
>
> We will add additional analysis. Our experiments suggest VLMs’ poor performance stems from selective perception: when comparing two objects composed of noncontiguous shapes, all frontier models improved by ~30% despite identical setups (L212–213). This gap shows that models only inspect items closely when they are perceived as distinct entities.
>
> To overcome this, a task could explicitly treat each object as a distinct entity—e.g., “a group of birds” rather than “flock”—to encourage active comparison. We believe reinforcement learning or chain-of-thought prompting would enable this behaviour in all settings, though even in the best “Unconnected” trials, model accuracy remains well below human levels.
>
> Moreover, our stepwise experiments (see “Break down the tasks” above) reveal that model performance degrades far faster than independent error multiplication suggests. Prompting models to execute visual steps separately may curb this error amplification, similar to the success seen in chain-of-thought  (Wei et al., 2022).
>
>
> ## o3 Results are missing. What is the reason?
>
> o3 was the most expensive model we evaluated, and per-trial it was about as expensive as all of the other models combined. However, OpenAI reduced the cost per-token on June 10th, and so we present the full results for o3 (with previous results repeated):
>
> Object Re-Identification, Standard: 55.2%
>
> Object Re-Identification, Unconnected: 79.2%
>
> Object Re-Identification, Pixel Perfect: 69.60%
>
> Visual Scavenger Hunt (VSH), Chain Length 2: 22.40%
>
> VSH, Chain Length 3: 18.4% (typo in Fig. 4, which says 18.5)
>
> VSH, Chain Length 4:  13.60%
>
> Circuit Connections, Single Color: 20%
>
> Circuit Connections, Standard: 32%
>
> Circuit Connections, Unique Colors: 38.4%
>
> o3 follows a similar performance trend across trials as Gemini 2.5, and o4-mini, bolstering our paper’s results. Interestingly, o3 and Claude 3.7— both slightly older models— do worse on VSH. This indicates newer frontier models have made progress on saccadic search.
>
> ## In VSH, is the model aware that some text points to non-existent objects? Do models self-correct?
>
> The models are not aware that the text points to non-existent objects and we did not see any instances of self correction. We examined the first twenty responses of all models across the chain length 3 variant, and we did not see a single instance of self-correction(L193-195). The low-performing models imitate the sentence structure of the one-shot example given but completely hallucinate a path and show no awareness that the shapes they mentioned do not exist on the board. The higher performance models had a much higher success rate of matching text-labels with shapes, but we did not observe a single instance of backtracking.
>
> However, these models would frequently say they could not find the next object on the board and give a random guess, such as the following output by Gemini 2.5 where it could not find the next shape (a magenta circle):
> ‘We need to find a "magenta circle". Looking at all the shapes, there is no circle that is magenta. This suggests a possible error in the label. One interpretation is that the label is a typo for "magenta square.’ It then proceeded to continue by checking the magenta square.
>
> In our initial inspection of the twenty first responses per model, we did not observe any instances of models explicitly saying they could not find a shape that truly did not exist. However, a search for the word “exist” revealed the following response by o4-mini after it landed on a cyan triangle by mistake:
> “We start at the cyan triangle. The text-label on this shape is "green circle". There are two greenish circles, a lime one and a darker green one. The path from the dark green circle leads to a dead end, as it instructs us to find an "orange triangle", which does not exist. Therefore, we must follow the path to the lime circle.” The model justifies its choices instead of backtracking.
>
> This suggests that the lack of self-correction is a fundamental part of model failure modes in VSH. We have edited the discussion around Fig. 5 to include this analysis.
>
>
>
> ## Can you extensively verify which tasks are feasible for breaking into small parts and solving and which are not?
>
> We posit that VSH is the most naturally decomposable task, and perform an additional experiment to verify its performance in this setting. We also add this analysis to the paper.
>
> We ran an additional experiment on VSH that breaks down saccadic search. Instead of asking the model to step on its own through the whole chain of length 3, we asked it to go one step at a time (also in a one-shot setting) with a chain length of 1. We did this for three steps, using the model’s answer as the starting point for the next chain.
> These are the results:
> o4-mini: 90.67%
> Gemini 2.5: 88%
> Llama 3.2 Vl (11b): 24%
> Qwen 2.5 VL 7B: 14.67%
> Qwen 2.5 VL 32B: 16.00%
>
> This experiment also allows us to capture the per-step error rate of each model. We report only the final-step error rate—calculated over trials in which the second‑to‑last step was answered correctly—since intermediate steps require specifying both the next shape and its color. Gemini-2.5 has a final-step error rate of 8% and o4-mini has an error rate of just 6%, whereas the other models all have an error rate above 85%.
>
> Gemini 2.5 and o4-mini’s higher performance in this setting shows that they are capable of the atomic unit of the saccadic search, but are just incapable of an autonomous, sustained saccadic search across the image. After all, simple error multiplication suggests this would yield a success rate of 83% for o4-mini and 77% for Gemini 2.5, which neither reach.
>
>  The lower performance of the other models is supported by prior work such as VLMs are blind, and suggests that they cannot perform the atomic unit of the task. We have included this analysis in the paper.
>
> It is not obvious to us how to break down line-tracing and object comparison into steps. Psychological research generally confirms that these are visual primitives (Ullman’s theory of visual routines).
>
>
>
> ## What approach is the model taking to solve in Circuit Crossings(4.3)?
>
> We hypothesize, following Wei et al.’s “Slow Perception” work, that the best‑performing models execute a series of discrete “jumps” along the wire, whereas the poorest performers attempt only a single global guess. Our log-odds analysis in Table 1 corroborates this: models show no evidence of smooth, human‑like contour tracing. We will revise the analysis section accordingly.
> Across all Circuit Connections trials, no model demonstrates true contour tracing. Models fall into two clusters. The first group—including Molmo (7B), Llama‑3.2 Vision (11B), etc—performs at or below the 14.3% random baseline on Standard, Single Color, and Unique Colors (Fig. 6). Their accuracy correlates only with wire proximity (distance effect) when they exceed chance in Standard or Unique Colors, and they show no sensitivity to crossings (p > 0.1). This indicates reliance on simple localization heuristics rather than actual tracing.
>
> The second group—Gemini 2.5 Pro, Claude 3.7 Sonnet, O4‑mini, and O3—outperforms random chance, but do not score highly enough on any variant to conclude they can perform contour tracing. When color cues are removed in the Single Color variant, accuracy plummets (e.g., Gemini 2.5 Pro from 47.9% to 27%). The Single Colors is designed to be the only trial where line-tracing is the only working strategy, as in other variants wire colors are often unique. This gap in performance demonstrates an inability to maintain a continuous follow‑the‑line process. The only piece of evidence that suggests that these models do try and trace lines is that they exhibit significant negative crossing coefficients in Single Color (Gemini 2.5 Pro β = –0.535, p = 0.024; O4‑mini β = –0.715, p = 0.031; Claude 3.7 Sonnet β = –0.931, p = 0.010). Intuitively, the task of line-tracing is hardest when two lines interfere, but should not interfere with color-matching heuristics. Only in these models is there a significant effect. However, because of their poor performance, we hypothesize their tracing capabilities consist of a heuristic of discrete, saccadic hops that “get stuck” at ambiguous crossings. This pattern matches Wei et al.’s “Slow Perception” hypothesis: rather than tracing continuously, they perform coarse jumps informed by color and proximity, failing where such cues conflict or vanish.

---

> > ### Comment · Reviewer_3qvb · 2025-08-06
> > **Thanks for the clarifications and additional results.**
> >
> > The discussion below has clarified some of my concerns and I have updated my score accordingly

---

### Official Review · Reviewer_wW7w · 2025-06-24

**Clarity:** 3
**Significance:** 4
**Originality:** 3
**Rating:** 5
**Confidence:** 4

**Summary:**

the authors present a 3-task benchmark of visual challenges easy for humans but potentially difficult for machines.
these consist of procedurally generated images spanning 3 categories: object re-identification up to rigid transformation, a visual scavenger hunt, and circuit connection following. The authors argue all of these tasks require "non-local" visual reasoning, i.e., instead of
The authors verify that the tasks are easy for humans (~100% in all cases), and easy to objectively evaluate the accuracy of.
But, all models, including frontier models/reasoning models/etc., fall significantly short.
Thus, the released datasets/environments form a new challenge for vision+language models.

**Questions:**

my wishes for this work, in question form:

- did you try few-shot benchmarks?
- did you try fine-tuning?
and maybe most of all:
- (if fine-tuning or another method works:) does patching this type of reasoning with synthetic images translate into improvements on tasks requiring this type of reasoning on photographic images?

**Ethical Concerns:**

["NO or VERY MINOR ethics concerns only"]

**Final Justification:**

I maintain my positive rating, the authors added the requested experiments and strengthened their work.

**Limitations:**

somewhat; I think even the obvious parts of my wishlist (e.g., fine-tuning/few-shot) were not addressed in the limitations section. I think the marginal risk of this work for society is low given that the images are all synthetic constructions.

**Quality:**

3

**Strengths And Weaknesses:**

Strengths:

I quite liked this paper! It's difficult to find simple tasks where models fail, and the authors have specified three of them. It's hard to argue against.

- I liked that the authors constructed a few versions of the tasks to attempt to isolate/specify which parts are particularly hard. This provides a nice easy-to-hard curriculum, which could inform training.
- The argument of the paper is very clear/convincing, e.g., human performance is given for all tasks.

Weaknesses:

- I worry that some of the prompts used to specify the tasks are underdetermined. For circuit connections, while I used commonsense to infer that the breadboard is the big gray thing in the middle of the image and the components are the smaller squares with C_#; to make the task even less debatable, it would have been helpful to more precisely specify this either with text, or with a label in the image. A similar concern with the visual scavenger hunt: I didn't know what it meant to "follow the label" initially --- it would have been helpful to be more precise, like: "starting at the teal triangle, read the text ontop of it, and then "follow the label," i.e., move to the shape described by that label. Repeat the process of following the label for 3 steps, in total."

- It would have been nice to have some few-shot results in addition to the zero-shot results.

- While I appreciate the sentiment, the authors claim in L126 that the task is purely visual and cannot be simulated in natural language. I don't know if "ascii art" constitutes natural language, but I could imagine rendering all of the author's challenges (save for the visual scavenger hunt which relies on color) in that format. (in fact, that would be a potentially cool ablation of whether or not this task is fundamentally bottlenecked by the visual encoder itself).

Would have liked:

- I would have liked to have seen a non-synthetic-image version of these tasks --- while not programatically generate-able, you could imagine annotating versions of these tasks on some photographic images.

- I am curious to see if training on synthetic data like this (whether via sft or RL in an environment: both are supported by the author's work) will "patch" this issue, and whether it will generalize to photographic images.


Overall, my assessment is positive, but, I would have liked to have seen some of the ideas taken one step further, e.g., with fine-tuning as a pointer as to if this problem is going to be very hard to solve or somewhat patchable with enough post-training data.

---

> ### Author Rebuttal · Authors · 2025-07-31
>
> Thank you for your insightful review! We are encouraged that you found the paper convincing. We have addressed your feedback below.
>
> ## “It would have been nice to have some few-shot results in addition to the zero-shot results.”
> All tasks and models in the paper were evaluated in a few-shot setting(L191). This was a core part of our experimental setup to mitigate any prompt ambiguity and solely test the tasks themselves.
>
> However, for completeness of this response, we evaluated a subset of models in a zero-shot setting as well.  All of these examples were generated independently for each model. These results are not comparable to each other and with the results in the paper (Fig.s 2,4,6), as the examples were generated (via random sampling) per-model, but they are comparable between few-shot and zero-shot. They also have different difficulty levels(such as a reduced maximum number of wires from fifteen to twelve, and a different minimum rotation amount in Object-Reidentification). However, the results between a single model between few-shot and zero-shot are comparable:
>
> ### Object Re-Identification
> Standard, zero-shot: Claude 3.7 Sonnet (48.00 %), Gemini 2.5 Pro (57.40 %), o4‑mini (58.20 %), o3 (54.23 %), Molmo 7B (49.20 %), Llama 3.2 VL (11B) (49.00 %), Qwen 2.5 VL (7B) (46.00 %)
>
> Standard, one-shot: Claude 3.7 Sonnet (56.00 %), Gemini 2.5 Pro (54.33 %), o4‑mini (65.60 %), o3 (57.50 %), Molmo 7B (48.00 %), Llama 3.2 VL (11B) (49.00 %), Qwen 2.5 VL (7B) (47.40 %)
>
> ### Visual Scavenger Hunt (VSH)
> Chain Length = 3, zero-shot: Gemini 2.5 Pro (50.00 %), o4‑mini (32.00 %), Qwen 2.5 VL (7B) (12.80 %)
>
> Chain Length = 3, one-shot: Gemini 2.5 Pro (40.00 %), o4‑mini (34.00 %), Qwen 2.5 VL (7B) (13.2%)
>
> ### Circuit Connections
> Standard Variant, zero-shot: Claude 3.7 Sonnet (42.67 %), Gemini 2.5 Pro (52.00 %), o4‑mini (56.00 %), Llama 3.2 VL (11B) (18.20 %), Qwen 2.5 VL (7B) (12.80 %), Molmo 7B (20.60 %),
>
> Standard Variant, one-shot: Claude 3.7 Sonnet (40.40 %), Gemini 2.5 Pro (52.00 %), o4‑mini (50.00 %), Llama 3.2 VL (11B) (17.40 %), Qwen 2.5 VL (7B) (13.20 %), Molmo 7B (18.40 %).
>
> Because the performance between zero-shot and one-shot does not differ that much, using few-shot should also not make a big difference. However, we note that o4-mini and Claude 3.7 Sonnet may be an exception on Object-Reidentification, where their performance increases. This indicates that there is some small amount of ambiguity that is cleared up with the examples.
>
> ## The authors claim in L126 that [circuit connections] is purely visual and cannot be simulated in natural language.
>
> We do not claim this but we agree that the original working in the paper "This task is purely visual and cannot be effectively simulated in natural language." can be misleading and will revise it accordingly. What we meant to say is that the way that humans solve this task cannot be simulated in natural language. While the humanlike algorithm for Object Re-Identification and VSH can be intuitively broken down into natural language statements— “The red triangle was scaled up, so they are not the same object…” or “First we go to the red triangle, and then we see it say blue square…”— contour tracing has no intermediate steps in natural language in the way that humans perform it. It can only be broken down into smaller instances of line-tracing. Psychological research, such as Ullman’s theory of visual routines, confirms the idea that line-tracing is a primitive visual technique for humans.
> We agree with the reviewer that these tasks could be represented non-visually, such as through ASCII art, and think this is a good direction for future work.
>
> ## "I would have liked to have seen a non-synthetic-image version of these tasks”
>
> We construct a more natural version of VSH and evaluate models on it.  by taking images from the COCO dataset and rendering labels on the objects which point to other objects in the scene. We use image segmentation data to find the pixels that belong unambiguously to one object and render text labels at those locations. We evaluate several models on this in a two-shot setting with the following prompt (the starting object is different per-image):
>
> "Starting at the cat in the image, follow the text labels for 2 steps. (For instance, in a different example you might start at the topmost car, then go to the leftmost person, then the rightmost bottle. The answer would be bottle.) After those steps, what object are you on? Answer with the object name in curly braces, e.g. {car}."
>
> These are the results:
>
> Claude 3.7 Sonnet: 33%
>
> Gemini 2.5 Pro: 38.67%
>
> Qwen 2.5 VL 7b: 26.67%
>
> Qwen 2.5 VL 32b: 40.00%
>
> Llama 3.2 11b: 28%
>
> Gemma 3 27b: 40%
>
> These results show that the smaller models are much more performant on the natural version of the task. However, it is important to note that this task is very different from the shape-based version— it may have between 3 and 10 text annotations instead of the consistent 25 in the synthetic version of VSH, but the average number is four.
>
> Random accuracy is also not well-defined because what is an object in real-life is not well-defined, but a simplified version of random accuracy is picking a label at random which leads to 25% accuracy. Notably, we do not observe reasoning chains among the two models which did poorly on the synthetic variants but did better on this one- Gemma 3 (32b) and Qwen 2.5 (32b). We suspect this is because it is not a true visual search— image context gives strong hints about where objects are. We suspect these models use their internal representations of the image to model the short chain without actually performing saccadic search. Gemini 2.5 and Claude 3.7, in contrast, show saccadic traces on almost every example. We hypothesize this is more difficult, as there are many more spots of ambiguity to jump to on a natural image.
>
> ## "I am curious if training on synthetic data like this will "patch" this issue, and whether it will generalize to photographic images."
> We finetune two models on our tasks and find that only Visual Scavenger Hunt can be patched with finetuning. We used a peak learning rate of 1e-4, AdamW optimizer and a cosine learning rate schedule. We trained on each task for a single epoch of 20,000 images.
>
> After being trained, only Qwen7b on VSH was able to achieve 100% accuracy on a separate validation set. Furthermore, it was trained on chain lengths of 3 and 4, and achieved only 20.8% accuracy on chains of length 2 and 22% accuracy on chains of length 5.
>
> For Object Re-Identification, the Qwen 2.5 (7B) reached 77% accuracy and Llama 3.2 (11B) reached 84% accuracy on the Unconnected variant. For Circuit Connections, Qwen 2.5 (7B) reached 77.6% accuracy and Llama 3.2 (11B) reached 70.80% accuracy on the Standard variant.
>
> Preliminary testing of these fine-tuned models on the semi-natural task (see “non-synthetic” section above) was inconclusive; the model output was degenerate. More research is needed to see if reinforcement learning can solve this problem without causing catastrophic forgetting.
>
> We did not conduct a hyper-parameter search to try to reach higher accuracy. However, these models were tested and trained on data from the same variants. Based on these fine-tuning experiments and the results of frontier models in Fig. 6, we believe it is unlikely that existing architectures are capable of smooth visual search. However, we do not feel that there is enough evidence to say that current VLMs architectures cannot be trained to perform comparative perception and saccadic search.
>
> ## The Circuit Connections prompt is not fully determined
>
> The Circuit Connections task is determined because we write in the full prompt that the breadboard is the grey rectangle. The prompt in Fig. 1 is a truncated version of the full prompt, which can be found in Fig. 5 of the supplemental. We also provide the model with few-shot solved examples, which it can use to determine what the components are.
>
> ## The Visual Scavenger Hunt(VSH) prompt is underdetermined
>
> The VSH task is not undetermined because we included a single solved example with a fully annotated path (e.g., "We start at the red triangle, which says blue square. We then go to the blue square.... Finally, we end on the {red circle}") before asking the model to solve a new example. This detailed annotation defines "follow the label,"as the label-shape path is unique in the image given. The full prompt can be found in Fig. 4 of the supplemental.

---

> ### Comment · Reviewer_wW7w · 2025-08-05
> **thank you!**
>
> I maintained my already very positive score based on these responses and additional experiments. A pointer to Fig 4 in the supplemental would be great to include in the caption of the figure which introduces the task.

---

> > ### Author Response · Authors · 2025-08-06
> > **Thank you**
> >
> > Thank you. We will fix it in the final paper.

---

### Official Review · Reviewer_94Tc · 2025-07-01

**Clarity:** 3
**Significance:** 3
**Originality:** 2
**Rating:** 4
**Confidence:** 3

**Summary:**

Based on the observation that human reasoning about images often involves comparing parts of the image that are spatially separated, the authors devise three evaluation tests for VLMs. The three tests are loosely inspired by human visual behaviors, i.e. “comparative perception”, a form of visual search (“saccadic search”), and curve tracing (“smooth visual search”). To this end, the authors provide three procedurally-generated task evaluations, Object Re-Identification, Visual Scavenger Hunt, and Circuit Connections, which are supposedly straightforward for humans, and evaluate 12 current VLMs. The results show that these visual tasks are still challenging for current VLMs.

**Questions:**

Could the behavior of the models described in the section “Failure Modes” be made more quantitative? It is difficult to understand what exactly is meant by “in contrast, higher-performing models possess higher visual capacity but only utilize it selectively. They do not perform well when objects are presented as coherent, connected entities. These models have strong implicit priors about entity analysis.”
or similarly
“However, their vision is either too fuzzy to perform the same tracing when wire colors are shared or they do not perform tracing.”

The authors state that “the object is always kept in-frame and unoccluded”, but Figure 1 suggests that objects are occluded. Could you clarify?

How unique are the present tasks compared to other VLM benchmarks involving reasoning on diagrams, including ARC, concept ARC, and Ravens progressive matrices?

**Ethical Concerns:**

["NO or VERY MINOR ethics concerns only"]

**Final Justification:**

I acknowledge the solid answers and additional evaluations by the reviewers and will maintain my increased score, although I am still not convinced that this is a fundamental contribution.

**Limitations:**

yes

**Quality:**

3

**Strengths And Weaknesses:**

The manuscript is reasonably clearly written, and the figures illustrate the tasks and evaluation results well.

Evaluating the capabilities of VLMs is currently a topic of interest to the ML community.

While the three tasks clearly seem to capture specific properties of visual tasks that are difficult for current VLMs, it is not clear how important it is to devise an additional benchmark for VLM. As an example, the classic Bongard problems contain images requiring “saccadic search” or “smooth visual search”.

Regarding 3.3, the authors are encouraged to cite relevant literature, such as the “visual routines” idea by Shimon Ullman.

Minor & typos:
Our structured evaluation suite allows us if VLMs can perform similar visual
algorithms to humans.
specific environments environments
Please reformulate: “We manually observe the first 20 responses of all models”
… “with high confidence (p≤ 0.05) across almost models on the Standard and Unique Colors trials.”

---

> ### Author Rebuttal · Authors · 2025-07-31
>
> We appreciate your review and request to motivate our work against existing benchmarks! We have addressed your feedback below.
> ## How important is it to devise an additional benchmark  if Bongard problems contain images requiring these skills?
>
> Our benchmark is important because it is a targeted evaluation of visual reasoning skills; other benchmarks do not tell us if VLMs fail specifically because they lack these skills. Bongard problems require puzzle solving ability, and failing on them doesn’t mean you can’t perform visual algorithms. We  isolate and test visual algorithms like line tracing and object comparison.
>
> These visual algorithms are important because without robust low‐level visual reasoning, we cannot guarantee that models are using visual evidence instead of making assumptions about the image in front of them. Even the small VLMs we test solve ChartQA at above 80% accuracy and score above 60% on MathVista. Yet their vision is demonstrably brittle: they score below 20% on BlindTest, and frontier models score much worse on ChartQaPro than on ChartQa by 30–60%. This suggests that VLMs lean on pretrained world priors to fill gaps in what they cannot analyze visually (as also shown in Vo et al.’s “VLMs are Biased,” 2025).
>
> ## “The authors are encouraged to cite relevant literature, such as the “visual routines” idea by Shimon Ullman.”
>
> We agree and will add psychological studies into our paper to ground our evaluation. We will cite Ullman’s visual routines framework to explain how our saccadic search task mirrors his postulated “scan” and “trace” routines when following discrete targets and continuous contours, and we will reference Treisman’s Feature Integration Theory to clarify how the VLMs are unable to compare object features under some circumstances.
>
> ## “Minor & typos: “Our structured evaluation suite allows us if VLMs can perform similar visual algorithms to humans.” “specific environments environments” Please reformulate: “We manually observe the first 20 responses of all models” … “with high confidence (p≤ 0.05) across almost models on the Standard and Unique Colors trials.””
>
> We will reformulate these to be clearer and error free.
>
> ## Could the behavior of the models described in the section “Failure Modes” be made more quantitative?
>
> Yes, we will edit the failure mode section to be more quantitative. For object Re-ID, we will discuss the models’ higher accuracy on Unconnected than Connected, and how this means they process contiguous shapes differently. For Visual Scavenger Hunt, we will edit the paper to discuss the single-step performance of the models across the trials. For Circuit Connections, we will discuss how our log-odds analysis allows us to say which models are possibly performing tracing and hypothesize about what each model is doing.
>
> ### “It is difficult to understand “in contrast, higher-performing models possess higher visual capacity… They do not perform well when objects are presented as coherent, connected entities….”
>
> We will revise our text to make it more clear.  What we wanted to say is that because frontier VLMs score higher on the Unconnected variant than on the Standard variant, this means that they perform worse when objects are connected entities. We will edit this analysis into the paper.
>
> Fig. 3’s shows that O4‑mini, Gemini 2.5 Pro, and Claude 3.7 Sonnet demonstrate moderate object‑distinction skills—but only in specific scenarios. On the Standard variant, their accuracy is near random chance (56%, 53%, and 49% respectively, as shown in Fig. 2). However, their performance significantly improves on both the Unconnected (O4-mini: 74%, …) and Pixel-Perfect (O4-mini: 50.4%, …) variants. The only difference is that in the Standard variant, the component shapes of the object are always physically contiguous (L141).
>
> The models’ low accuracy on physically connected elements shows they fail to carefully inspect connected entities. We will also edit our paper to connect this to Gestalt principles of grouping (e.g., by proximity and connectedness) and how they generally make connected objects easier to process and compare for humans.
>
> ### It is hard to understand “However, their vision is either too fuzzy to perform the same tracing when wire colors are shared or they do not perform tracing.”
>
> We will revise our text to make it more clear. We wanted to say that because the top performing VLMs we tested doubled their accuracy from a setup in which all wire colors were the same to one in which each wire color was different, they are not capable of performing actual tracing.
>
> Gemini 2.5 Pro, Claude 3.7 Sonnet, and O4-mini perform better than other models but still demonstrate limited tracing abilities. Their accuracy drops significantly when color cues are removed or made uniform While tracing is expected to be harder when wires of the same color interfere, this low performance suggests either a lack of, or severely limited, tracing ability.
>
> What distinguishes these models from the others is that they show a statistically significant negative crossing effect in the Single Color trial, where wires only cross other identically colored wires. Gemini 2.5 Pro shows a log-odds coefficient of -0.5349 (p=0.0244) for the crossings effect in this variant. This behavior is consistent with a tracing algorithm, because crossings are visually ambiguous points. However, because these models also do poorly in Unique Colors as compared to humans, we suspect these models employ a higher-granularity version of the "visual localization" heuristic, possibly performing saccadic movements to find successive points instead of smoothly tracing it.
>
> ## The paper states ’the object is always kept in-frame and unoccluded’, but Fig. 1 suggests that objects are occluded.
>
> The object is not occluded because we define the object as the configuration of shapes in Image 1 and do not consider the components of the object occluding each other as the object being occluded. The primary object in our task is composed of component shapes—such as triangles and rectangles—which together form a single object. These components may partially occlude each other, similar to how the intersecting lines in an “X” overlap. However, such self-occlusion is considered intrinsic to the object and does not violate our definition of occlusion.
>
> Image 1 is literally just the primary object itself so it is trivially unoccluded. In Image 2, where we add distractor objects, we guarantee that no part of the object is ever occluded by a distractor object. In negative examples where the object is transformed, we do not guarantee that the component shapes won’t occlude each other in Image 2.
>
> ## How unique is this compared to ARC, Ravens progressive matrices(RPMs), etc?
>
> Our benchmark is unique compared to these because we evaluate lower level skills in isolation. Unlike Bongard, ARC, and RPM puzzles, our tasks isolate visual algorithms from abstract reasoning. Solving Bongard problems forces you to think about a conceptual rule across two example sets and even know the rule’s name; ARC puzzles demand hypothesis testing on limited examples to extrapolate abstract patterns; and Raven’s matrices rely on relational reasoning to complete interactive patterns. These are puzzles that can be difficult for humans to solve. Our task strips away the reasoning and the knowledge. Our problems are intuitive— there is no puzzle to solve, and the human solve rate is >=99%. We specifically target primitive visual algorithms, such as holding objects in working memory (comparative perception) or continuous visual tracing (smooth visual search).
>
> Solving Bongard problems does not imply mastery of the basic visual reasoning we identified—searching over images or tracing lines—nor does failing one imply incapacity, since Bongard tasks demand puzzle‑solving skills. The skills we test are very primitive, and to our knowledge no prior evaluation has isolated them. New models like o3, Gemini 2.5 Pro, and Claude 3.7 perform much better on “VLMs Are Blind,” and our central research question is whether those gains translate into consistent execution of the low‑level visual algorithms humans use.

---

> > ### Comment · Reviewer_94Tc · 2025-08-05
> >
> > I acknowledge the solid answers and additional evaluations by the reviewers. While I am still not convinced that this is a fundamental contribution, I will increase my scores.

---

### Decision · Program_Chairs · 2025-09-17

**Decision:**

Accept (spotlight)

**Comment:**

The paper received four expert reviews. The authors provided a rebuttal that attempted to address the concerns raised in the reviews. The reviewers read the rebuttal and engaged with the authors.  The reviewers unanimously like the paper and recommended accept. The area chair agreed with the recommendation and decided to accept the paper. Congratulations! Please see the reviews for feedback on the paper to revise the final version of your paper and include any items promised in your rebuttal.